# Psychological Distress Associated with Enforced Hospital Isolation Due to COVID-19 during the “Flatten the Curve” Phase in Morocco: A Single-Center Cross-Sectional Study

**DOI:** 10.3390/healthcare12050548

**Published:** 2024-02-26

**Authors:** Sarra Chadli, Rhita Nechba Bennis, Naoufel Madani, ElMostafa El Fahime, Redouane Abouqal, Jihane Belayachi

**Affiliations:** 1Acute Medical Unit, Ibn Sina University Hospital, Rabat 10056, Morocco; sarrachadli@gmail.com (S.C.); r.abouqal@um5r.ac.ma (R.A.); 2Laboratory of Biostatistics, Clinical, and Epidemiological Research, Faculty of Medicine and Pharmacy, Mohammed V University of Rabat, Rabat 10100, Morocco; 3Molecular Biology and Functional Genomics Platform, National Center for Scientific and Technical Research (CNRST), Rabat 10000, Morocco; 4Neuroscience and Neurogenetics Research Team, Faculty of Medicine and Pharmacy, Mohammed V University of Rabat, Rabat 10100, Morocco

**Keywords:** mental health, hospital isolation, anxiety, depression, HADS, COVID-19, low–middle-income countries, Morocco

## Abstract

(1) Background: although much research has highlighted the mental health challenges faced by patients in hospital isolation during the COVID-19 pandemic, data from low–middle-income countries, including Morocco, are lacking. The main objective of this study was to assess the psychological distress of patients undergoing enforced hospital isolation during the initial phase of the COVID-19 pandemic in Morocco. (2) Methods: we conducted a cross-sectional study between 1 April and 1 May 2020, among patients hospitalized in isolation for suspected or confirmed COVID-19 at the Ibn Sina University Hospital of Rabat, Morocco. Anxiety and depression were assessed using the Hospital Anxiety and Depression Scale (HADS). Binary logistic regression was performed to identify variables associated with anxiety and depression, with a cutoff of ≥8 used for both scales to create dichotomous variables. (3) Results: among 200 patients, 42.5% and 43% scored above the cut-off points for anxiety and depression, respectively. Multiple logistic regression identified female gender, a higher education level, a longer duration of isolation, and a poor understanding of the reasons for isolation as significant factors associated with anxiety. Conversely, female gender, chronic disease, a longer duration of isolation, and a poor understanding of the reasons for isolation were factors significantly associated with depression. (4) Conclusions: our study underscores high rates of anxiety and depression among patients forced into hospital isolation during the initial phase of COVID-19 in Morocco. We identified several factors associated with patients experiencing psychological distress that may inform future discussions on mental health and psychiatric crisis management.

## 1. Introduction

The coronavirus disease 2019 (COVID-19) pandemic caused a global health crisis on many levels. While considerable research has investigated the mental health outcomes of patients in hospital isolation during COVID-19, most findings are issued from high-income countries (HICs) [1,2,3,4,5]. In contrast, due to limited data, low–middle-income countries (LMICs) received less attention in policy dialogues [6,7]. However, the evidence available, although scarce, showed an increased vulnerability of LMIC populations to the negative mental impact of the pandemic due to high rates of poverty, less comprehensive mental health and social care systems, and limited mental health services. Investigators have pointed out that psychological distress is more acute and more consequential in LMIC societies, warranting further investigations to inform policy responses to future waves and future pandemics [8,9,10].

In this context, Morocco is an important country in which to examine psychological distress, as its strong ties with Europe led it to be the first and one of the most prominent African nations affected by COVID-19. After recording its first case on 2 March 2020, the country entered the first phase of the pandemic, known as the “flatten the curve” phase, which aimed to slow the spread of the virus and prevent the overwhelming of healthcare systems [11]. As of 20 March 2020, the Moroccan government officially declared a health state of emergency (SOE), enabling it to enact laws and regulations to restrict people’s movements and activities. Although it was initially set for three months, the SOE was extended several times, up to 10 June 2021, lasting more than 14 months [12,13].

Like many governments, Morocco imposed a total lockdown to control the spread of the virus. In adherence to the Moroccan government’s directives during the SOE, all individuals suspected of COVID-19 were placed in hospital isolation in single rooms until the results of their PCR tests were available. Every confirmed or highly suspected case was consistently kept in hospital isolation during the viral phase, for a minimum of 7 to 10 days, without any visits allowed from relatives. After discharge, the patients were confined for an additional two weeks until their PCR returned negative. When proper isolation at home was not possible, individuals were placed in single hotel rooms, under strict surveillance from the authorities.

The psychological impact of isolation has been the focus of much investigation. Previous studies have reported the negative mental health consequences of isolation and quarantine among the general population and healthcare workers, such as anxiety and depression, fear, post-traumatic stress disorder, eating disorders, and addictive or suicidal behaviors [14]. In Morocco, national surveys revealed that anxiety and depression were significantly prevalent among the general population during confinement [15,16,17]. This was also assessed in specific groups, including children and adolescents, medical students, and healthcare professionals [18,19,20,21,22,23]. However, little data are available regarding patients in enforced hospital isolation for COVID-19 in the North African country [24].

To address this critical research gap, we conducted a comprehensive cross-sectional study to investigate the relationship between enforced hospital isolation and patients’ mental health during the first phase of the pandemic in Morocco. At the beginning of our study on 1 April 2020, the total number of COVID-19 cases reported in Morocco was 654, with 39 deaths, out of a legal population of 36,236,242 [25].

The main objective of our work was to evaluate the prevalence of anxiety and depression in this population using the Hospital Anxiety and Depression Scale (HADS), and secondly, to identify the various factors associated with patients experiencing these symptoms while in hospital isolation during the initial phase of COVID-19.

## 2. Materials and Methods

### 2.1. Study Design and Settings

This cross-sectional, monocentric study was conducted in an isolation unit at the Ibn Sina University Hospital Center of Rabat, Morocco, and was carried out from 1 April 2020 to 1 May 2020.

Consecutive patients aged over 18 years old and hospitalized in isolation for suspected or confirmed COVID-19 were included. Patients who died, who were unable to understand the questions or communicate, and those who did not wish to participate were excluded.

### 2.2. Data Collection

The sociodemographic characteristics, medical characteristics, and characteristics related to the healthcare system and environment were recorded. To evaluate symptoms of anxiety and depression among isolated patients during their hospital stay, we used the HADS. The questionnaires were delivered either by telephone or in person. The survey took approximately 20 min to complete.

#### 2.2.1. Sociodemographic and Medical Characteristics

Epidemiological data included age, gender, marital status (married or not), academic education (none, primary school, secondary school, or post-graduate), and professional status (student, unemployed, employed, or retired). The patients’ history comprised chronic disease and toxic use (tobacco, alcohol, drugs). A history of traumatic events was noted, defined as exposure to actual or threatened death, serious injury, or sexual violence by directly experiencing it, witnessing it, learning that it had occurred to a relative, or being exposed to extreme or repeated aversive details of it (DSM V criteria).

The initial symptomatology was classified into respiratory, digestive, neurological, and general symptoms. Upon admission, the patients’ vital signs were recorded. The results of the chest CT scan and SARS-CoV-2 RT-PCR test from the nasopharyngeal swab were noted. Therapeutic management was also documented.

#### 2.2.2. Adherence to Public Health Authorities’ Directives

We surveyed patients about their compliance with the preventive measures recommended by the Public Health Authorities during the pandemic, including confinement at home, maintaining a social distancing of at least 1 m, mask wearing in public, frequent handwashing or the use of hand sanitizer, avoiding touching the face with hands, coughing or sneezing into a single-use tissue or the sleeve of the elbow or the upper arm, immediate dispose of used tissue papers, room ventilation, and the avoidance of sick people. We categorized the patients’ adherence to these nine measures into four levels: poor (less than 4 measures applied), moderate (4 to 6 measures applied), good (6 to 8 measures applied), and very good (9 out of 9 measures applied). The patients also rated their self-perception levels of understanding of the disease, understanding of the reasons for isolation, and adherence to the infection control directives, using a scale from 0 (worst self-perception) to 10 (best self-perception).

#### 2.2.3. Hospital Isolation and Communication

According to the Moroccan health authorities, a COVID-19 suspected case was defined as an individual having clinical criteria for acute respiratory infection along with epidemiological criteria of close contact with a confirmed case [26]. Irrespective of their clinical conditions, all suspected individuals were forced into hospital for isolation in single rooms until the results of their PCR tests were available. Patients with negative tests and low evidence of infection were discharged, while patients with negative tests and high evidence of infection (context of close contact, clinical symptoms, and radiological features) were continued to be isolated in single hospital rooms. Regardless of their clinical state and need for hospital care, all patients with positive tests were hospitalized in isolation, either in a single room or a shared room (with a maximum of three other positive patients), depending on hospital bed availability. In all cases, isolation was maintained for a minimum of 7 to 10 days, covering the viral phase.

For each patient, we documented the type of accommodation (single or shared room) and the length of the hospital stay (per day). The patients’ phone disposal, network access, and social media use were specified. We noted their major sources of information concerning the outbreak and categorized them as public health authorities, health care providers, media, regular websites, and word of mouth. Furthermore, we asked patients to indicate the average number of daily rounds from the medical and paramedical staff, and to rate, using a scale from 0 to 10, their quality of communication with healthcare providers and their satisfaction with the medical care provided.

### 2.3. Outcome Measures

Symptoms of anxiety and depression among patients during their hospital stay were evaluated using the HADS. This validated instrument is made of 14 items, seven related to anxiety symptoms and seven related to depression symptoms (Appendix A). Each item is rated from 0 to 3, and two scores are obtained for both the anxiety (HADS-A) and depression (HADS-D) dimensions, with a maximum score of 21 each [27]. The prevalence of anxiety and depression was assessed using a score of 8 as a cut-off value. The Arabic version of this tool was deemed appropriate for our study without any cultural adaptation [28,29].

### 2.4. Statistical Analysis

Continuous variables were reported as mean and standard deviation for variables with normal distributions and as median and interquartile range (IQR) for variables with skewed distributions. The normality of the distribution was tested using the Kolmogorov–Smirnov test with Lilliefors correction. Categorical variables were presented as percentages within each category. Group comparisons were conducted using Pearson’s chi-squared test and a linear-model ANOVA. To evaluate the internal consistency of the HADS items, Cronbach’s coefficient alpha was employed. A high alpha coefficient (≥0.70) suggests that the items within a scale measure the same construct, supporting the construct validity. Anxiety and depression were considered the dependent variables.

Binary logistic regression was performed to identify variables associated with anxiety and depression. Variables were selected based on their clinical relevance and a literature review [1,3,24,30,31,32,33,34]. For each explanatory variable, odds ratios (ORs), 95% confidence intervals (CIs), and *p*-values were reported. A cutoff of ≥8 was used separately for anxiety and depression scales to create dichotomous variables. A two-tailed *p*-value < 0.05 was considered statistically significant. Statistical analyses were conducted using Jamovi 2.4 and the R software version 4.3.2.

## 3. Results

### 3.1. Sociodemographic and Medical Characteristics

A total of 200 participants were included. The study population consisted entirely of Moroccan individuals, with a mean age of 40 ± 15 years (18–78). The gender distribution revealed a male-to-female ratio of 1.5, with men accounting for 61% of the patients. In terms of matrimonial status, 52% of the patients were married. Educational attainment varied among the participants, with 50% having completed post-graduate education, 36% with a secondary degree, and 14% with elementary education or no formal education. The majority of patients were employed (69%), including 15% who were healthcare professionals, while 14% were unemployed, 11% were retired, and 10% were students. Past history included chronic disease (32%) and toxic use (23%). A history of traumatic events was noted in 16.5% of the cases.

Patients manifested respiratory (47%), general (47%), neurological (30.5%), and digestive (27%) symptoms. Upon admission, a mean of 20% of the patients presented with desaturation in room air, requiring oxygen support. COVID-19 was confirmed via RT-PCR tests for 132 patients (66%), while 34% had negative testing and remained in isolation as highly suspected cases. In the chest CT scan, the pulmonary extent was classified as moderate to severe in 53.5% of the cases. The patients were mainly treated with hydroxychloroquine (91%), azithromycin (95%), anticoagulants (88%), glucocorticoids (68%), and antibiotics (55%).

### 3.2. Adherence to Public Health Authorities’ Directives

The study participants assessed their adherence to preventive measures recommended by Public Health Authorities as poor (14.5%), moderate (22.5%), good (42.5%), and very good (20.5%). Using a self-perception scale from 0 to 10, the patients’ mean levels of understanding of the disease, understanding of the reasons for isolation, and adherence to the infection control directives were reported as 7.6, 8.9, and 9.4, respectively.

### 3.3. Hospital Isolation and Communication

In 68% of the cases, the patients were isolated in single rooms, while 32% of them were placed in shared rooms. The average duration of isolation was 8 ± 3 days, with a maximum duration of 25 days. During their hospital stay, nearly all patients had a mobile phone (99.5%), access to a network (93%), and social media use (89.5%). Their major sources of information regarding the COVID-19 outbreak were Public Health Authorities 78.5%), media (70.5%), healthcare providers (44%), regular websites (24.5), and word of mouth (19%). The patients stated to have received an average of five visits per day from the medical and paramedical staff. They evaluated their quality of communication with healthcare providers and their satisfaction with medical care provided as means of 8.9 and 9.0, respectively.

The descriptive characteristics are summarized in Table 1.

### 3.4. Hospital Anxiety and Depression Scale (HADS)

According to the HADS, 85 patients scored above the cut-off value for anxiety (HADS-A) and 86 patients scored above the cut-off value for depression. Based on these scores, the prevalence of anxiety and depression in the patient sample was calculated to be 42.5% and 43%, respectively. A univariable analysis was performed to identify which covariates were more related to anxiety and depression (Table 1).

A subsequent multiple logistic regression indicated that the factors associated with anxiety were female gender (OR = 2.54; 95% CI (1.34; 4.78) *p* = 0.004), secondary or university level (OR = 3.55; 95% CI (1.08; 11.64) *p* = 0.03), longer duration of isolation (OR = 1.10; 95% CI (1.01; 1.20) *p* = 0.01), and poor understanding of the reasons for isolation (OR = 0.71; 95% CI (0.55; 0.92) *p* = 0.01).

On the other hand, the factors associated with depression were female gender (OR = 3.53; 95% CI (1.82; 6.84) *p* < 0.001), chronic disease (OR = 3.22; 95% CI (1.43; 7.24) *p* = 0.005), longer duration of isolation (OR = 1.09; 95% CI (0.99; 1.19) *p* = 0.05), and poor understanding of the reasons for isolation (OR = 0.67; 95% CI (0.52; 0.87) *p* = 0.003)

The results of the multiple logistic regression are presented in Table 2.

## 4. Discussion

This research aimed to assess the mental health status of patients who were forced into hospital isolation during the initial phase of the COVID-19 pandemic in Morocco. Among the 200 patients enrolled, we observed that 42.5% and 43% experienced high levels of anxiety and depression, respectively. In the multiple logistic regression, female gender, a higher education level, a longer duration of isolation, and a poor understanding of the reasons for isolation were statistically associated with anxiety. Conversely, the factors associated with depression were female gender, having a chronic disease, a longer duration of isolation, and a poor understanding of the reasons for isolation.

The high rates of anxiety and depression observed within our group of patients align with the results reported in studies from other countries, such as Ethiopia and China, which documented comparable levels of anxiety and depression linked to hospital isolation during the first phase of the pandemic [30,31]. In Morocco, an online survey conducted among the general population during the outbreak yielded similar results, with 49% and 53% of respondents reporting anxiety and depression, respectively [15].

The sociodemographic characteristics showed that female patients were more likely to experience negative mood alterations and higher levels of anxiety, which is consistent with the findings of previous research [33,35]. It is well known that the female gender is associated with a higher risk of psychological distress, even considered by some authors to be a predictive factor for developing mental disturbances [36,37,38]. This gender gap in emotional well-being is suggested to be related to the great hormonal fluctuations in women, which have the potential to influence neurotransmitters and neurosteroids and impact behavioral processes [37].

Furthermore, we noted that individuals with higher education levels exhibited higher levels of anxiety, as reported by other countries during the pandemic [32,33,39]. This association can be attributed to the enhanced understanding of the severity of the disease and the huge uncertainty around its various consequences, inducing heightened fear and concern. Previous surveys also evidenced high levels of fear of the novel coronavirus, mostly due to risk for both oneself and loved ones, an intolerance of uncertainty, and massive media exposure [40]. Similarly, during the outbreak of the Middle East Respiratory Syndrome (MERS), individuals were more likely to have negative psychological manifestations over fear [41].

Notably, the presence of a chronic disease demonstrated a high correlation with patients experiencing feelings of depression, which supports the results of studies conducted in Turkey and Iran [32,34]. This can be explained by past experiences of health adversities, leading to an increased perception of life threat. A cross-sectional study in Ethiopia further illustrates this, revealing that hospitalized COVID-19 patients with medium and high perceived life threat were three and five times more likely to have psychological distress [30].

Our findings also showed that the length of isolation was correlated with patients’ mental health. In a case–control study comparing patients in hospital isolation for an infectious disease with patients who did not require isolation, anxiety and depression symptoms worsened after one week of isolation and continued to be significant after the second follow-up [42].

Additionally, patients who had a poor understanding of the reasons for their isolation were found to be less accepting of the situation and more likely to experience anxiety and depression. This finding was echoed by Gammon et al., who reported that most complaints of isolated patients were related to a lack of information and poor communication. The positive effect of information has been recognized for over 30 years, supported by several studies affirming its efficacy in reducing patients’ anxiety [41,42,43]. This was also highlighted in the context of hospital isolation during the initial phase of COVID-19, with researchers underscoring the imperative need for accurate knowledge regarding isolation in order to improve patients’ emotional and behavioral adjustment [31,33].

Overall, we found high levels of anxiety and depression among patients forced into hospital isolation during the COVID-19 pandemic in Morocco. The relevant factors identified to be significantly associated with patients experiencing psychological distress were female gender, a higher educational level, chronic disease, a longer duration of hospitalization, and a poor understanding of the reasons for isolation.

## 5. Practical Implications

Based on our findings, we recommend minimizing the duration of isolation whenever possible. Implementing common rooms for positive patients, providing various materials (TV, books, cards, board games, etc.), and facilitating communication with relatives, can help to reduce feelings of boredom and loneliness. Patients should receive comprehensive written and verbal information about the reasons for isolation, as well as regular updates on their management and care. Hence, adequate training of healthcare professionals is crucial to ensure effective communication, which includes delivering clear explanations to patients, checking their understanding, and engaging in active listening. Identifying high-risk patients is of paramount importance for early prevention through specialized psychological interventions, and ensuring appropriate follow-up when necessary.

## 6. Limitations

This study included 200 patients, which may not be representative of the entire population of COVID-19 patients hospitalized in isolation in Morocco. Nevertheless, within the context of the early phase of the pandemic, the sample size was significant, as only 654 COVID-19 cases were recorded in Morocco at the start of the study [7]. This research was conducted in a single-center tertiary hospital, which may also be a limitation to the generalizability of the results. As this was a cross-sectional exploratory study, we could not deduce any causal relationship or inform on the long-term effects of anxiety and depression in these patients.

## 7. Conclusions

Our study highlights high levels of anxiety and depression among patients forced into hospital isolation in Morocco during the initial phase of the COVID-19 pandemic. We identified female gender, a higher education level, chronic disease, a longer duration of isolation, and a poor understanding of the reasons for isolation as significant factors associated with psychological distress. To tackle these challenges, global health strategies should prioritize patients with preparatory and accurate information regarding isolation, enhance effective communication with healthcare professionals, shorten the length of isolation, and implement measures to reduce boredom and loneliness. The early identification of individuals at high risk for anxiety and depression is crucial, allowing for timely and targeted interventions to prevent psychological distress. These findings offer valuable insights to support patients’ mental well-being during future public health emergencies, not only in Morocco, but also in other countries facing similar challenges.

## Figures and Tables

**Table 1 healthcare-12-00548-t001:** Descriptive characteristics and univariable analysis of the factors associated with anxiety and depression among COVID-19 patients in enforced hospital isolation at Ibn Sina University Hospital from 1 April to 1 May 2020.

		Anxiety			Depression		
	*N* = 200	No Anxiety (*n* = 115)	Anxiety (*n* = 85)	*p*-Value	No Depression (*n* = 114)	Depression (*n* = 86)	*p*-Value
Age ** (years)	41 (15)	41 (15)	40.5 (16)	0.89 ^1^	41 (14)	40 (16)	0.52 ^1^
Gender *				0.009 ^2^			<0.001 ^2^
Male	122 (61)	79 (69)	43 (51)		82 (72)	40 (46.5)	
Female	78 (39)	36 (31)	42 (49)		32 (28)	46 (53.5)	
Marital status *				0.52 ^2^			0.05 ^2^
Non-married	96 (48)	53 (46)	43 (51)		48 (42)	48 (56)	
Married	104 (52)	62 (54)	42 (49)		66 (58)	38 (44)	
Academic education *				0.35 ^2^			0.65 ^2^
None–primary level	26 (13)	16 (14)	10 (12)		15 (13)	11 (13)	
Secondary level	74 (37)	45 (39)	29 (34)		44 (39)	30 (35)	
University level	100 (50)	54 (47)	46 (54)		55 (48)	45 (52)	
Professional status *				0.56 ^2^			0.07 ^2^
Unemployed	30 (15)	16 (14)	14 (16.5)		12 (10.5)	18 (21)	
Student	21 (10.5)	11 (9.5)	10 (12)		10 (9)	11 (13)	
Employed	138 (69)	82 (71)	56 (66)		88 (77)	50 (58)	
Retired	11 (5.5)	6 (5)	5 (6)		4 (3.5)	7 (8)	
Chronic disease *				0.19 ^2^			0.01 ^2^
No	63 (31.5)	83 (72)	54 (63.5)		86 (75)	51 (59)	
Yes	137 (68.5)	32 (28)	31 (36.5)		28 (25)	35 (41)	
Toxic use *				0.40 ^2^			0.94 ^2^
No	154 (77)	91 (79)	63 (74)		88 (77)	66 (77)	
Yes	46 (23)	24 (21)	22 (26)		26 (23)	20 (23)	
History of traumatic events *				<0.001 ^2^			0.009 ^2^
No	167 (83.5)	108 (94)	59 (70)		102 (89.5)	65 (76)	
Yes	33 (16.5)	7 (6)	26 (31)		12 (10.5)	21 (24)	
Level of application of the preventive measures *				0.25 ^2^			0.42 ^2^
Poor	29 (14.5)	18 (16)	11 (13)		19 (17)	10 (11.6)	
Moderate	45 (22.5)	20 (17)	25 (29)		18 (16)	27 (31.4)	
Good	85 (42.5)	49 (43)	36 (42)		51 (45)	34 (39.5)	
Very good	41 (20.5)	28 (24)	13 (15)		26 (23)	15 (17)	
Results of PCR test *				0.78 ^2^			0.40 ^2^
Negative	68 (34)	40 (35)	28 (33)		36 (32)	32 (37)	
Positive	132 (66)	75 (65)	57 (67)		78 (68)	54 (63)	
Duration of isolation (day) **	8 (3)	1 (1)	1 (1)	0.14 ^1^	1.3 (1)	1.3 (1)	0.62 ^1^
Type of room *				0.07 ^2^			0.02 ^2^
Single	137 (68.5)	73 (63.5)	64 (75)		71 (62)	66 (77)	
Shared	63 (31.5)	42 (36.5)	21 (25)		43 (38)	20 (23)	
Number of staff visits (day) **	5 ± 2	5 (1.5)	5 (1)	0.39 ^1^	5 (1.5)	5 (1)	0.68 ^1^
Major sources of information							
Public health authorities *				0.42 ^2^			0.85 ^2^
No	43 (21.5)	27 (23.5)	16 (19)		24 (21)	19 (22)	
Yes	157 (78.5)	88 (76.5)	69 (81)		90 (79)	67 (78)	
Media *				0.33 ^2^			0.61 ^2^
No	59 (29.5)	37 (32)	22 (26)		32 (28)	27 (31)	
Yes	141 (70.5)	78 (68)	63 (74)		82 (72)	59 (69)	
Regular websites *				0.16 ^2^			0.75 ^2^
No	151 (75.5)	91 (79)	60 (70)		87 (76)	64 (74)	
Yes	49 (24.5)	24 (21)	25 (29)		27 (24)	22 (26)	
Health care providers *				0.68 ^2^			0.23 ^2^
No	112 (56)	63 (55)	49 (58)		68 (60)	44 (51)	
Yes	88 (44)	52 (45)	36 (42)		46 (40)	42 (49)	
Word of mouth *				0.07 ^2^			0.33 ^2^
No	162 (81)	98 (85)	64 (75)		95 (83)	67 (78)	
Yes	38 (19)	17 (15)	21 (25)		19 (17)	19 (22)	
Communication level **	8.5 (2)	9 (1.5)	8 (2)	0.11 ^1^	8 (2)	8.5 (2)	0.51 ^1^
Satisfaction level **	9 (1)	9 (1)	9 (1.5)	0.12 ^1^	9 (1.5)	9 (1)	0.31 ^1^
Disease understanding level **	8 (2)	8 (2)	7 (2)	0.10 ^1^	8 (2)	7.5 (2)	0.15 ^1^
Isolation understanding level **	9 (1)	9 (1)	9 (1)	0.02 ^1^	9 (1)	9 (1.5)	0.004 ^1^
Adherence to infection control directives level **	9 (1)	9.5 (1)	9 (1)	0.28 ^1^	9.5 (1)	9 (1)	0.14 ^1^

* *N* (%), ** mean (SD); ^1^: linear-model ANOVA; ^2^: Pearson’s Chi-squared test.

**Table 2 healthcare-12-00548-t002:** Multivariable analysis of the factors associated with anxiety and depression among COVID-19 patients in enforced hospital isolation at Ibn Sina University Hospital from April 1st to May 1st, 2020.

	Anxiety			Depression		
	OR	95%CI	*p*-Value	OR	95%CI	*p*-Value
Age	0.98	0.96; 1.01	0.35	0.97	0.94; 1.00	0.09
Gender						
Female/Male	2.54	1.34; 4.78	0.004	3.53	1.82; 6.84	<0.001
Marital status						
Married/Non-married	1.02	0.52; 2.01	0.94	0.69	0.34; 1.41	0.31
Academic education						
None/Primary school	2.49	0.78; 7.92	0.12	2.47	0.74; 8.22	0.14
Secondary school/University	3.55	1.08; 11.64	0.03	3.09	0.90; 10.55	0.07
Chronic disease						
Yes/No	1.70	0.79; 3.64	0.17	3.22	1.43; 7.24	0.005
Type of room						
Single/Shared	1.84	0.87; 3.88	0.10	1.99	0.92; 4.33	0.08
Duration of isolation	1.10	1.01; 1.20	0.01	1.09	0.99; 1.19	0.05
Isolation understanding level	0.71	0.55; 0.92	0.01	0.67	0.52; 0.87	0.003

OR: odds ratio; CI: confidence interval.

## Data Availability

Data are contained within the article and Appendix A.

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
