# Peer review of "Psychological Distress Associated with Enforced Hospital Isolation Due to COVID-19 during the “Flatten the Curve” Phase in Morocco: A Single-Center Cross-Sectional Study"

_healthcare, 2024, doi:10.3390/healthcare12050548_

Round 1
Reviewer 1 Report
Comments and Suggestions for Authors
Thank you for inviting me to review this work. The purpose of this paper is to assess the psychological distress of patients undergoing enforced hospital isolation during the first phase of the pandemic in Morocco. The topic could be important in the year of 2020 and/or 2021, however, now as entering 2024 – the post-COVID-19 era, this study has fewer contributions to the new knowledge body and/or practical implications. There are four major drawbacks of this study: (1) this study surveyed a limited sample and used linear regressions, thus the results were not enough for reference for the current practices; (2) the discussion should more focus on how the findings can inform preparing future pandemic or make policy/practical recommendations on helping isolated patients to reduce depressions and anxiety; (3) conducting regressions, the authors can not conclude any causal inference, however, the authors stated “However, little is known about the specific effects of enforced hospital isolation during the COVID-19 pandemic in low-middle-income countries such as Morocco.” (Line 57-58); (4) in fact, there are lots of existing literature on enforced isolation and depressions and anxiety (Fancourt et al., 2021; Knopf, 2020; Loades et al., 2020, etc.), thus, the authors should review the existing literature more carefully, and accurately present what has found and what has not found. More detailed literature reviews should be added.
Minor issues:
1. No structured abstract is needed.
2. The survey participated patients were isolated, so how the authors delivered the questionnaires in person (lines 84-85) should be further clarified.
3. Please clarify the study surveyed gender or sex.
Reviewer 2 Report
Comments and Suggestions for Authors
The title should refer only to a single hospital in Morocco, not the general population because the study is a monocenter and included a limited sample size of 200 patients.
The abstract should be rewritten because it is impossible to understand what was done regarding statistical analysis. The conclusions are poor and not linked to the main results "we identified several risk factors" though the authors failed to explain which were such factors.
The introduction seems okay but needs to be reinforced with similar results from other populations or contexts assessed during the COVID-19 pandemic.
Authors should include a flow diagram that describes the overall process to gather their final sample size of patients included in the study. The methods sections must be separated into headings to better understand the methodological framework, data gathering, and analyses.
In my opinion, the statistical analysis approach is wrong regarding the use of simple and multiple linear regression because despite the anxiety and depression scores might be linear, all other variables were included in the model as dummy variables. However, the coefficients for them were not properly explained in the results as these were only cited as simple numbers. This is the major flaw of the study.
I recommend that instead of using simple and multiple LR, the authors should reanalyze their data using a multiple logistic regression using a binary outcome (depressed or not, anxious or not) in which the explaining variables are also entered as binary or multinominal. Instead of using the simple linear regression approach, authors should use the X2 test to find significant variables or instead use a variable selection approach.
When using a multiple logistic regression approach, the output from the models will give the authors OR values that could help them explain the contribution of risk factors in terms of the chance of being depressed or anxious in comparison to persons who were not.
Reviewer 3 Report
Comments and Suggestions for Authors
Thanks for the opportunity to evaluate this manuscript. It is valuable to assess the psychological distress of patients undergoing enforced hospital isolation. However, the study can have some further improvements:
1 The value of the research needs to be clearer. What is the difference from previous research? It is recommended that the author add some review on previous studies.
2 For the methods section, it is recommended that the author add sub- itle to describe the process, measurements, samples, analysis etc.
3 How are qualitative variable variables handled in multivariable linear analysis? Is T-test or ANOVA more suitable to explain the differences in depression and anxiety due to gender and other qualitative factors?
4 Some abbreviations in the table need to be noted, such as CI.
5 The conclusion section suggests that the author should not only describe the results, but also consider the differences with previous studies or any new findings. In addition, the relevant practical implications can also be further discussed.
6 It is recommended that the authors discuss research limitations and future studies separately.
Round 2
Reviewer 1 Report
Comments and Suggestions for Authors
Thank you for the revised manuscript. This revision addressed majority of my concerns. However, there are still some minor places that should be addressed before the acceptance. Please see below.
1. Line 43-45, while considerable research…so the authors need to point out specific research.
2. Still, since the authors can not draw causal inferences due to the limit of their research method, the authors should not use terms such as “effect”, “impact”, etc. For example, line 86, I would say “investigate the relationship between the enforced hospital isolation and mental health”. Please also check this issue throughout the manuscript.
Reviewer 2 Report
Comments and Suggestions for Authors
The authors have adequately addressed all the comments and suggestions. The change in the statistical approach enhanced the presentation of results and facilitated their interpretation as well as improved the conclusions drawn from the results. The authors have put effort into this revised version and now the manuscript can be accepted for publication in the journal.
Author Response
Dear Reviewer,
We want to express our sincere gratitude for your thorough review and thoughtful feedback on our manuscript. Your constructive comments have greatly contributed to improving our work, and we are deeply grateful for the insights and expertise you invested in this evaluation. Thank you once again for your invaluable assistance.
Best regards
Reviewer 3 Report
Comments and Suggestions for Authors
The manuscript has been significantly improved with revisions and can be considered for publication.
Author Response
Dear Reviewer,
Thank you once again for your review. Your insightful and constructive remarks have been invaluable in enhancing the quality of our work, and we are sincerely appreciative of the time and expertise you dedicated to this process.
Best regards